# Unveiling Primary Cutaneous B-Cell Lymphomas: New Insights into Diagnosis and Treatment Strategies

**DOI:** 10.3390/cancers17071202

**Published:** 2025-04-01

**Authors:** Zachary R. Barbati, Yann Charli-Joseph

**Affiliations:** 1Department of Pathology and Laboratory Medicine, University of California San Francisco, San Francisco, CA 94107, USA; zachary.barbati@ucsf.edu; 2Dermatology and Dermatopathology Private Practice, Mexico City 01090, Mexico

**Keywords:** primary cutaneous lymphomas, cutaneous lymphomas, B-cell lymphomas, cutaneous hematopathology

## Abstract

This review discusses the diagnosis and management of B-cell lymphomas of the skin, which can be primary (originating in the skin) or secondary (spreading from other areas, often lymph nodes). The focus is on the three key subtypes of primary cutaneous B-cell lymphomas: follicle center lymphoma and marginal zone lymphoma, which are indolent with excellent prognoses, and diffuse large B-cell lymphoma, leg type, which is aggressive and has a high mortality rate. Each subtype has distinct clinical, microscopic, and molecular features that guide diagnosis and treatment. Indolent subtypes are treated with localized therapies like radiation or skin surgery, while aggressive subtypes require chemotherapy and/or immunotherapy. The authors highlight emerging treatments targeting cancer-specific molecules or enhancing the immune response, and discuss medical knowledge gaps that remain for these rare cancers. They also emphasize the need for further research and emphazise the importance for inclusive studies to improve outcomes and health equity for diverse patient populations.

## 1. Introduction

B-cell lymphomas (BCLs) of the skin represent a rare and heterogeneous group of extranodal non-Hodgkin lymphomas [1]. These neoplasms, albeit infrequent, are clinically significant in dermatology, oncology, hematology, and pathology, due to their diagnostic and therapeutic complexity [2,3]. Lymphomas of the skin are classified as primary cutaneous (PCL), when confined to the skin at diagnosis, and secondary cutaneous (SCL), when the skin is involved following dissemination from systemic or nodal lymphomas [4,5].

The incidence of PCBCL has increased over recent decades, now affecting approximately 4 per million individuals. PCBCLs account for roughly 25% of PCL and demonstrate a higher prevalence in male, non-Hispanic White individuals, and adults aged over 50 years [6,7]. Data from the Surveillance, Epidemiology and End Results (SEER) program indicate that the incidence rate of PCBCL is 3.5 per 1,000,000 person-years in non-Hispanic White individuals, 2.8 in Hispanic Whites, 1.9 in Asian/Pacific Islanders, and 1.5 in Black people [2].

PCBCL are currently classified into three major subtypes by both the 5th edition of the World Health Organization (5-WHO) Classification of Hematologic Neoplasms [4] and the 2022 International Consensus Classification (2022-ICC) of Mature Lymphoid Neoplasms [7]: primary cutaneous follicle center lymphoma (PCFCL), primary cutaneous marginal zone lymphoma or lymphoproliferative disorder (PCMZL/LPD), and primary cutaneous diffuse large B-cell lymphoma, leg type (PCDLBCL,LT) [4,7]. These subtypes are further categorized based on clinical behavior and prognosis, with PCFCL and PCMZL/LPD being low-grade/indolent, and PCDLBCL,LT classified as a high-grade/intermediate-to-aggressive neoplasm [1,5,6,8]. Notable differences with previous classifications include the categorization of MZL/LPD as a separate entity from the broad group of extranodal marginal zone lymphomas of mucosa-associated lymphoid tissue (MALT lymphoma), its downgrading to an LPD (in the 2022-ICC), and the recognition that PCFCL may be composed of large cells, and regardless, should not be categorized as PCDLBCL (even when occurring on the legs), with the diagnosis relying on its characteristic germinal center B-cell immunophenotype and defining molecular features [4,7,9].

Although not the focus of this review, secondary cutaneous B-cell lymphomas (SCBCLs) share numerous overlapping features with PCBCL, making it essential to differentiate between primary and secondary cutaneous involvement in all suspected PCBCL cases [10,11]. Diagnostic workup for PCBCL follows the tumor, node, metastasis (TNM) staging system endorsed by the International Society for Cutaneous Lymphomas (ISCLs) and the European Organization for the Research and Treatment of Cancer (EORTC) [12]. A thorough evaluation should include clinical assessment for B-symptoms and organ-specific signs. Additionally, blood count with differential, a metabolic panel with lactate dehydrogenase (LDH) levels, and imaging studies (e.g., PET/CT or CT-SCAN with intravenous contrast of the chest, abdomen, pelvis, and neck if applicable) are performed, except in cases of PCMZL/LPD, where they are unnecessary given its growing recognition as an indolent lymphoproliferative disorder. Bone marrow biopsy, peripheral blood flow cytometry, and protein electrophoresis (to rule out monoclonal gammopathy) are optional and of limited value in indolent/low-grade PCBCL (especially in PCMZL/LPD) but are indicated in patients with systemic symptoms, widespread disease, unexplained cytopenias or leukocytosis and are mandatory in patients with PCDLBC,LT, where additional screening for infections (hepatitis B and C, human immunodeficiency virus, latent tuberculosis) is paramount before initiating systemic therapy [7,11,12,13,14].

Herein, we aim to comprehensively examine PCBCLs, detailing their clinical, histopathological, immunohistochemical, and molecular/genetic features [Figure 1], whilst focusing the core of this review on sensible patient management. We also sought to integrate the latest research findings, current treatment approaches, and emerging therapeutic strategies, with the goal of improving diagnostic accuracy and treatment outcomes. Additionally, we identify knowledge gaps and propose future research directions to enhance the management of these complex lymphoid malignancies.

## 2. Indolent/Low-Grade Primary Cutaneous B-Cell Lymphomas/Lymphoproliferative Disorders

### 2.1. Primary Cutaneous Follicle Center Lymphoma (PCFCL)

Primary cutaneous follicle center lymphoma is the most common subtype of PCBCL in Caucasians, representing up to 30 to 50% of PCBCL cases, and approximately 10% of all PCL [1,2,5]. By contrast, data from Korea and Japan show that PCFCL represents the least common PCBCL, accounting for less than 4% of all PCL, underscoring the importance of ethnicity in the relative distribution of PCL [15,16]. PCFCL predominantly affects adults, with a median age at diagnosis ranging from 51 to 58 years, and exhibits a slight male predominance [5,6,17]. The prognosis is generally excellent, with a 5-year survival rate exceeding 95%; although remarkably rare, cases located on the legs have a strikingly lower 5-year survival rate of approximately 41% [14,17,18,19].

Clinically, PCFCL typically presents as a solitary erythematous to violaceous nodule (or a cluster of nodules), or as a plaque, commonly located on the head (particularly the scalp), neck, or trunk [Figure 2] [5,14,17,18,19,20,21]. Less common clinical variants include a miliary or agminated form presenting as multiple papules on the face, which may mimic acne, rosacea or folliculitis [22]. Another variant, historically referred to as “Crosti’s lymphoma” or “reticulohistiocytoma of the dorsum” is characterized by figurate annular concentric plaques with peripheral macules or papules on the trunk. Multifocal and/or widespread cutaneous involvement is uncommon (15%), as is ulceration (scarcely reported, chiefly in longstanding cases) [21,22,23]. Although PCFCL is rare in Black American patients, it typically presents at a younger age and is associated with more regional or generalized disease, including the presence of scattered flat plaques [24]. Recurrence occurs in 20–50% of cases, but does not significantly impact long-term outcomes. Extracutaneous involvement is uncommon, occurring in roughly 10% of cases [19,20,21,22,23,25].

PCFCL is a neoplasm of mature germinal center B-cells: predominantly small centrocytes and, to a lesser extent in most cases, centroblasts (which may be medium- or even large-sized) [4,7,9,26]. Tumor cells occupy the reticular dermis, with variable perivascular, periadnexal, and deep-dermal accentuation, with occasional extension into the subcutaneous tissue. The growth pattern may be follicular, follicular and diffuse, or purely diffuse. PCFCLs with a follicular or follicular and diffuse pattern show distorted follicular aggregates (supported by expanded dendritic cell meshworks), that are irregular in shape and size, and that often directly abut adjacent follicles. These follicles are often devoid of mantle zones and typically lack tingible body macrophages [Figure 3]. By contrast, PCFCL with a purely diffuse growth pattern, mostly seen in long-standing or larger lesions, is characterized by sheet-like pandermal proliferations of germinal center neoplastic cells, lacking distinct follicular structures and their supporting follicular dendritic cell (FDC) meshworks [4,5,6,14,21,23,26,27,28].

Immunohistochemically, neoplastic cells express pan-B cell markers, as well as germinal center markers such as B-cell lymphoma-6 (BCL6), stathmin-1, LIM-only transcription factor-2, human germinal center-associated lymphoma and activation-induced cytidine deaminase. The expression of cluster of differentiation (CD) 10 (CD10) is inconsistent. Neoplastic cells typically do not express the anti-apoptotic protein BCL2 (positive in 10–30% of cases). Additionally, lymphoma cells demonstrate minimal staining for interferon regulatory factor-4 (IRF4)/multiple myeloma oncogene-1 (MUM1), with nuclear expression being present in less than 30% of cells. The proliferative index assessed using Ki67/mindbomb homolog-1 (MIB1) is low, with fewer than 50% of neoplastic cells proliferating, compared to greater than 90% in reactive germinal centers. CD21, CD23, and CD35 staining highlight irregular or distorted FDC meshworks [Figure 4] [1,2,5,6,9,17,23,26,27,28].

Molecular analysis reveals clonal immunoglobulin heavy-chain (IgH) rearrangements in roughly 50% of cases through polymerase chain reaction (PCR), and with a higher yield with newer methods, such as tissue flow cytometry and next generation sequencing [1,5,28].

*BCL2*-*IGH* rearrangement [t(14:18)], characteristic of nodal follicular lymphoma (FL) is typically absent in PCFCL, even in cases that show BCL2 expression. In PCFCL, such overexpression is likely associated with chromosomal amplification of *BCL2* rather than with a translocation. Rarely, point mutations in *BCL6*, *RhoH*/*TTF*, and *PAX5*, resulting from aberrant somatic hypermutation, have been reported. Gene expression profiling (GEP) has demonstrated upregulation of genes associated with a germinal center B-cell-like pattern, including *SLAM*, *LCK*, and *SPINK2*. Additionally, overexpression of the oncogenic 17–92 microRNA (miR) cluster, particularly miR20a and miR-20b, has been linked to a subset of PCFCL cases and is posited to correlate with a worse prognosis [1,5,28,29,30,31,32,33,34,35].

### 2.2. Primary Cutaneous Marginal Zone Lymphoma/Lymphoproliferative Disorder (PCMZL/LPD)

Primary cutaneous marginal zone lymphoma/lymphoproliferative disorder accounts for approximately 20–40% of PCBCL and about 7% of all PCL. It prevails in middle-aged adults, with a median age at diagnosis of around 55 years, and demonstrates a male predominance, with a male-to-female ratio of 2:1 [1,2,5]. The prognosis is excellent, with a 5-year disease-specific survival of ≥98%, albeit a high rate of cutaneous recurrence is observed in roughly 50% of cases [12,36]. Extracutaneous spread occurs in fewer than 10% of cases [18,20].

PCMZL/LPD is distinct from extracutaneous mucosa-associated lymphoid tissue (MALT) lymphomas due to differences in its etiological factors, histopathologic features, immunoglobulin and chemokine receptor expression patterns, genetic translocations, and frequency of systemic dissemination [1,4,5,28]. Whilst PCMZL/LPD is still designated as a “lymphoma” in the latest iteration of the WHO Classification of Hematologic Neoplasms [4], it was downgraded to a “lymphoproliferative disorder” in the 2022 International Consensus Classification of Mature Lymphoid Neoplasms [5], owing to its extremely indolent clinical behavior and underscoring the importance of avoiding unnecessary staging studies (such as PET/CT and bone marrow biopsy) and aggressive treatments (such as intravenous rituximab or chemotherapy) in PCMZL/LPD [1,2,5].

The pathogenesis of PCMZL/LPD is thought to be linked to chronic antigenic stimulation, notably to *Borrelia burgdorferi* infection, although this association has been solely confirmed in a subset of patients from endemic areas in Europe. Furthermore, PCMZL/LPD is significantly associated with systemic conditions, including gastrointestinal disorders, autoimmune diseases, and other malignancies [1,5,28,36,37,38,39].

Clinically, PCMZL/LPD typically presents as multiple (less often solitary), slow-growing, erythematous to violaceous papules, small nodules, or plaques, predominantly occurring on the trunk and/or extremities, with sparing of the head and neck [Figure 5] [1,5,28,37]. Hypopigmented patches have also been reported in Black American patients [24]. While exceedingly rare, atypical variants include anetodermic and agminated forms, the latter resembling the agminated/miliary variant of PCFCL [39].

PCMZL/LPD is a neoplasm of mature post-germinal center B-cells, which may be outnumbered by admixed reactive T-cells. Neoplastic B-cells are small to medium-sized and have been variably referred to as “marginal zone”, “centrocyte-like” or “monocytoid” B-cells. Additionally, lymphoplasmacytoid cells and plasma cells are often present, particularly at the periphery of the infiltrate and within the superficial dermis. The histopathologic pattern in PCMZL/LPD, similarly to PCFCL, is typically nodular or diffuse, centered upon the reticular dermis, with sparing of the papillary dermis and epidermis [1,5,6,26,27,28,40]. Reactive germinal centers may also be observed, either with distinct mantle zones or, in well-established lesions, colonized by neoplastic cells, resulting in a loss of the typical demarcation between germinal centers and their surrounding mantle zones [Figure 6] [1,5,6,27,28].

Immunohistochemically, neoplastic B-cells express pan-B cell markers, with strong and diffuse co-expression of BCL2 [1,5,28]. Plasma cells may be highlighted by CD38, CD138 or MUM1, and exhibit monotypic cytoplasmatic immunoglobulin light-chain restriction (either kappa or lambda), with a kappa-to-lambda ratio of 5:1 to 10:1 [Figure 7] [5,10,27,28,41].

Two subtypes of PCMZL/LPD are currently recognized based largely on IgH expression: the more common class-switched form (approximately 90%), which expresses IgG (with 40% of cases expressing IgG4 in the absence of IgG4-related disease), and, to a lesser extent, IgA or IgE. The rarer non-class switched subtype expresses IgM, akin to non-cutaneous MALT lymphoma, and is associated with a higher rate of extracutaneous spread compared to the class-switched form, which does not appear to portend an unfavorable outcome [42,43,44]. Though still a matter of debate, due to the differences mentioned above, some experts advocate that the class-switched form should be designated as a “lymphoproliferative disorder” (LPD), while the non-class switched subtype may be considered as a bona fide yet completely indolent “lymphoma” [42,43,44].

Aberrant somatic hypermutation has been described in several genes, including *PAX5*, *RhoH/TTF*, *cMYC*, *PIM1*, *SLAMF1*, *SPEN*, and *NCOR2*. Notably, recent studies have identified *FAS* mutations in 63–68% of PCMZL/LPD cases, leading to impaired apoptosis of neoplastic cells [1,5,28,45,46,47].

### 2.3. Treatment of Indolent/Low-Grade Primary Cutaneous B-Cell Lymphomas/Lymphoproliferative Disorders

Given the generally favorable prognosis of indolent/low-grade PCBCL, a conservative therapeutic approach should always be prioritized when clinically appropriate [Figure 8] [1,5,9,17,18,19]. This may include a “wait and see” strategy for solitary PCMZL/LPD, which, while acceptable, is often not preferred by patients and necessitates close surveillance [30].

For both PCMZL/LPD and PCFCL with solitary or limited lesions, the treatment modalities of choice are typically involved-site radiation therapy (ISRT) and, less commonly, surgical excision. ISRT is often favored due to its lower morbidity and excellent outcomes, particularly in older adults, or in anatomic areas where surgical excision may result in suboptimal cosmetic outcomes or healing difficulties, such as the face and lower leg, respectively. While both surgical excision and ISRT can provide complete responses (CRs) in nearly all cases, relapses are common, and optical surgical margins remain ill-defined. Reported surgical margins are generally less than 1 cm, while ISRT typically employs margins ranging from 1 to 5 cm [1,5,30,40,41].

Regarding ISRT, there is no consensus on the optimal radiation doses for indolent PCBCL, with treatment regimens varying widely. Success has been reported from very low-dose ISRT (VLD-IRST; 2–4 Gy in 2 fractions) to standard-dose ISTR (SD-IRST; 24–40 Gy in 12–20 fractions). Both regimens have demonstrated similar efficacy (CRs), but VLD-ISRT is associated with a significantly lower incidence of side effects (15.7% vs. 78.4%, *p* < 0.0001), making it an attractive option for treating indolent PCBCL. Although some cohorts suggest a higher relapse rate following excisional surgery compared to ISRT, statistically significant differences have not been established [14,36,48,49,50,51,52,53]. Nevertheless, excision is typically only favored in select cases, where small lesions may be removed with minimal non-disfiguring surgery [9].

Alternative treatment strategies for solitary or limited lesions of indolent PCBCL include intralesional corticosteroids, particularly for small lesions of PCMZL/LPD, though the CR rate is suboptimal at approximately 44% and multiple treatment cycles are often required [54,55]. Systemic antibiotics such as cephalosporins and tetracyclines may be effective for lesions associated with *Borrelia burgdorferi* infection (especially in endemic regions of Europe) [56,57]. Other therapies include intralesional interferon-alpha (though not available in many countries) [58], and intralesional rituximab, a chimeric monoclonal antibody targeting CD20. Intralesional rituximab has shown high CR rates (60–80%) but is often impractical due to the need for multiple injections, associated pain, wheal-like reactions (both at the injection site and distant sites), and mild, albeit bothersome, adverse events, such as urticaria, exanthem, fever, nausea, and malaise. Importantly, rituximab can lead to rare, but potentially serious reductions in B-cell counts, even with low cumulative doses [59,60,61].

In patients with multiple disseminated lesions, or in select cases (i.e., PCFCL with extensive scalp lesions, where IRST may lead to long-standing alopecia), intravenous rituximab (325 mg/m^2^ weekly for 4–8 infusions) has proven effective, with reported 98% overall response rates (ORRs), 64% CRs, a median progression free survival (PFS) of 58 months, and a median time to next treatment (TTNT) of 60 months in the largest series of 25 cases of PCMZL/LPD. In a similar cohort of 29 patients with PCFCL, systemic rituximab achieved an ORR of 96%, a CR of 72%, a median PFS of 78 months, and a median TTNT of 85 months. In both cohorts, adverse events, mostly grade 1–2, occurred in approximately 32–38% of cases [62,63,64,65,66].

Chemotherapy for indolent/low-grade PCBCL has failed to show reduced relapse rates compared to less aggressive therapies and should only be considered as a last-resort option for patients with extensive cutaneous involvement, or extracutaneous spread, and in PCFCL involving the lower extremities, which carries an unfavorable prognosis [1,5,6,28,38]. For PCMZL/LPD with widespread lesions, a few European studies have evaluated chlorambucil as single-agent chemotherapy, achieving CR rates greater than 60% [5,14,28]. Multi-agent chemotherapy regimens are almost universally discouraged, given the potential for significant side effects [1,5,9,28,67]. However, rare cohorts of multi-agent chemotherapy in PCMZL/LPD have demonstrated CR rates of approximately 85%, though relapses occur at similar rates to those observed with other treatment modalities. Similarly, a systematic review of multi-agent chemotherapy in PCFCLs found CR rates in 85% of patients, with a relapse rate of 44%. The cyclophosphamide, doxorubicin, vincristine, and prednisone (CHOP) regimen has been the most commonly used chemotherapy protocol, often with or without the addition of rituximab and/or ISRT. However, experience with these combined approaches remains limited by the relatively small patient populations treated with such regimens [5,28,67].

### 2.4. Differential Diagnosis for Indolent/Low-Grade Primary Cutaneous B-Cell Lymphomas/Lymphoproliferative Disorders

PCMZL/LPD and PCFCL are both characterized by infiltrates of small neoplastic B-cells, often accompanied by varying numbers of reactive T-cells. This combination of small cell size and mixed immunophenotype entails a myriad of differential diagnoses, including both reactive and neoplastic conditions, which must be accurately identified due to significant variations in management and prognosis [10,28].

The most common diagnostic challenges involve distinguishing between PCMZL/LPD and PCFCL, as well as differentiating these from cutaneous B-cell hyperplasia, commonly referred to as cutaneous lymphoid hyperplasia (CLH) or “reactive B-cell rich lymphoid proliferation” under the 5-WHO classification [3,27,28].

In the differential diagnosis of neoplastic conditions, one notable entity is primary cutaneous small/medium CD4-positive T-cell lymphoproliferative disorder (PCSMCD4LPD), which shares many similarities, especially with solitary PCMZL/LPD [3,10,27,28,68,69,70,71,72,73]. A critical differential diagnosis for PCFCL is secondary cutaneous involvement by nodal or systemic follicular lymphoma (FL), which shares almost identical histopathological and immunophenotypic features with PCFCL. The primary distinguishing feature is the almost ubiquitous expression of BCL2 in FL due to the characteristic (14;18) (*IGH::BCL2*) translocation, which is less commonly observed in PCFCL (~17% of cases). Overall, BCL2 expression can be observed in up to 30% of PCFCL cases. While BCL-2 expression in PCFCL has not been shown to confer a more aggressive disease course, it is associated with a higher risk of cutaneous relapse [31,74]. Thus, regardless of BCL2 expression, a thorough staging evaluation is essential to rule out FL in cases of suspected PCFCL [1,3,5,10,27,28].

Other nodal or systemic neoplasms of mature small B-cells that may occasionally involve the skin should be considered the differential diagnosis of indolent/low-grade PCBCL, notably chronic lymphocytic leukemia (CLL) and mantle cell lymphoma (MCL), an exceedingly rare but aggressive B-cell malignancy [3,10,27,28]. An in-depth analysis of other less common secondary cutaneous B-cell neoplasms falls beyond the scope of this review, but such conditions have been previously detailed in a publication by one of the authors herein [10].

## 3. Intermediate to Aggressive/High-Grade Primary Cutaneous B-Cell Lymphomas

### 3.1. Primary Cutaneous Diffuse Large B-Cell Lymphoma, Leg Type (PCDLBCL,LT)

Primary cutaneous diffuse large B-cell lymphoma, leg type (PCDLBCL,LT), first described in 1996, is an aggressive subtype of PCBCL that accounts for 1–4% of all PCL [75,76,77]. This entity predominantly affects older adults, with a median age at diagnosis of approximately 70 years, and exhibits a marked female predominance, with a female-to-male ratio ranging from 2:1 to 3:1 [1,2,25]. PCDLBCL,LT has an aggressive clinical course; despite a high rate of initial CR rates following standard treatment, relapses occur in roughly 70% of cases. The 5-year survival rate ranges from 40% to 60%, influenced by factors such as patient age, extent of tumor dissemination, protein expression patterns (particularly loss of P16 and presence of MYC expression exceeding 40%, both associated with a poor prognosis), and underlying molecular characteristics, including mutations in B-cell receptor signaling genes such as *CARD11* and *CD79A/B* [1,28,75,78,79].

Clinically, PCDLBCL,LT typically presents as rapidly enlarging erythematous to violaceous nodules or plaques, primarily located on the lower extremities. Although confined to the legs in most cases, 10–15% of patients may exhibit involvement of other regions, such as trunk or neck. Lesions are often unifocal (approximately one-third of cases), may ulcerate, and cause significant discomfort or pain [Figure 9] [1,5,25,27,28,78,80]. In Black patients, PCDLBCL,LT typically presents with hyperpigmented nodules that may mimic granulomatous lesions, such as those seen in sarcoidosis or infectious granulomas [24]. Constitutional B-symptoms are observed in approximately 10% of patients at diagnosis, with elevated LDH levels in around 12%. Extracutaneous dissemination, most commonly to the lymph nodes, bone marrow, and central nervous system, occurs in about 45% of cases [1,5,25,27,28].

Histopathologically, PCDLBCL,LT is marked by dense, diffuse sheets of large (≥3 times that of reactive lymphocytes), atypical lymphocytes (immunoblasts) infiltrating the dermis and often extending into the subcutis [Figure 10] [1,5,10,25,27,28,75,77].

Immunophenotypically, PCDLBCL,LT cells express pan-B cell markers, and habitually express BCL2, MUM1/IRF4 (unrelated to *IRF4* rearrangements), and FOXP1. BCL6 expression is variable and typically weak if present. Immunoglobulin expression is most often restricted to IgM. However, co-expression of IgD is present in approximately 50% of cases. MYC expression, unrelated to *MYC* rearrangements, is present in more than 55% of cases. A high mitotic rate is typically observed, with Ki-67 indices exceeding 50% and occasionally as high as 90% [Figure 11] [1,5,10,25,27,28].

PCDLBCL,LT demonstrates a post-germinal center B-cell phenotype with monoclonal IgH rearrangements observed in most cases. Lymphoma cells often harbor significant somatic hypermutations and retain superantigen binding sites, suggesting they underwent germinal center processes akin to those in the activated B-cell-like (ABC) subtype of nodal diffuse large B-cell lymphoma. Notably, one study demonstrated that lymphoma cells of PCDLBCL,LT exhibit defects in class switch recombination and express IgM without undergoing isotype switching, indicating a blockage at the transitional stage between germinal center B-cells and plasma cells [80,81].

PCDLBCL,LT has a distinct molecular profile compared to nodal DLBCL and shares similarities with other special-site DLBCL, such as those in the breast, testes, and central nervous system, which frequently harbor *MYD88* and *CD79b* mutations [82]. Gene expression profiling studies have revealed that *MYD88* L265P mutations are present in approximately 75% of PCDLBCL,LT cases, contributing to chronic activation of the nuclear factor-kappa B (NF-κB) pathway [81,82,83,84,85,86]. *MYD88* L265P mutations are associated with a shorter survival when compared to patients with wild-type *MYD88* [87]. Mutations in *CD79B* are frequently observed, enhancing B-cell receptor signaling and supporting tumor survival and proliferation [82,83]. Deletions or mutations of *CDKN2A* are present in roughly two-thirds of cases and are associated with a markedly reduced 5-year DSS of 43% vs. 70% when absent [82,83,84]. In PCDLBCL,LT, *MYC* rearrangements have been associated with shorter DSS and disease-free survival rates (without differences in OS) and occur at a higher incidence than in nodal DLBCL (32 vs. 9–14%) [86]. Per one study, co-expression of MYC and BCL2 (dual expressor status) can be identified in up to 83% of PCDLBCL,LT cases and is associated with a more aggressive disease course [88]. By contrast, double-hit statuses (involving rearrangements of *MYC* and either *BCL2* or *BCL6*) are occasionally observed but have not been shown to significantly impact prognosis [89].

### 3.2. Treatment of PCDLBCL,LT

The standard first-line treatment for PCDLBCL,LT involves polychemotherapy with the R-CHOP regimen. The addition of ISRT has been shown to enhance local control and prolong PFS, with a median PFS of 58 months when combined with R-CHOP, compared to 14 months with R-CHOP alone [9,78,79,90,91]. Importantly, localized treatment alone is typically inadequate for effective disease management [90]. In patients who are unable to tolerate the standard R-CHOP regimen due to advanced age or significant comorbidities, alternative reduced-intensity regimens or the combination of rituximab with pegylated liposomal doxorubicin (PLD) may be considered. The use of PLD is advantageous as it has a reduced risk of cardiotoxicity compared to conventional doxorubicin [Figure 8] [76].

Due to limited experience with the treatment of relapsed or refractory PCDLBCL,LT, the National Comprehensive Cancer Network (NCCN) guidelines recommend second-line therapies typically used for systemic DLBCL. These therapies now include chimeric antigen receptor (CAR) T-cell therapy, polatuzumab vedotin ± bendamustine ± rituximab, the CEOP regimen (cyclophosphamide, etoposide, prednisolone, and vincristine) ± rituximab, among others [9,91]. Limited evidence suggests potential benefits of lenalidomide and ibrutinib in the treatment of PCDLBCL,LT, with a single case report indicating the possible efficacy of polatuzumab vedotin [92,93,94,95]. Notably, data presented at the 2024 5th World Congress for Cutaneous Lymphomas showed promising results for patients receiving polatuzumab-based chemotherapy in combination with CAR T-cell therapy [96]. A Phase II clinical trial of patients with refractory or relapsed PCDLBCL,LT demonstrated a 26.3% ORR at 6 months following single-agent treatment with lenalidomide, with dose adjustments correlating with improved response rates [92,97]. Additionally, ibrutinib, a Bruton’s tyrosine kinase inhibitor, has demonstrated efficacy, particularly in cases with *MYD88* mutations, where it appears to offer significant therapeutic benefit [93,94]. Several clinical trials, including ZUMA-7 and BELINDA, have established the efficacy of CAR T-cell therapy as a second-line treatment for large B-cell lymphoma, with at least one reported case of PCDLBCL,LT in the ZUMA-7 trial [98,99,100]. These findings suggest a potential shift in the treatment paradigm for large B-cell lymphoma, which may have significant implications for the management of PCDLBCL,LT in the future.

Recent advances in diagnostic and monitoring techniques are critical for the management of PCDLBCL,LT. F-18 fluorodeoxyglucose PET/CT is increasingly used for both staging and assessing response to therapy. However, its long-term prognostic value remains to be fully validated in larger studies [101]. Additionally, circulating tumor DNA (ctDNA) analysis using digital droplet PCR (ddPCR) represents a non-invasive approach to detect minimal residual disease, offering promising utility for monitoring disease status, particularly in cases with *MYD88* mutations [102].

Although exceedingly rare, cases of spontaneous regression have been reported, suggesting possible immune-mediated mechanisms that are not yet fully understood [103]. These observations highlight the need for further exploration of immune-based therapies and of the tumor microenvironment in PCDLBCL,LT, potentially leading to novel therapeutic strategies.

### 3.3. Differential Diagnosis for PCDLBCL,LT

From a clinical and therapeutic standpoint, the primary differential diagnosis for PCDLBCL,LT is PCFCL with a diffuse growth pattern and a prominence of large centroblasts. PCDLBCL,LT typically expresses BCL2, MUM1/IRF4, FOXP1, and IgM and exhibits a high Ki-67 proliferation index. It often lacks CD10 and shows minimal or absent BCL6 expression. By contrast, PCFCL strongly expresses BCL6 (with less frequent CD10 positivity), and BCL2 expression is usually low or absent. Additionally, PCFCL demonstrates a significantly lower proliferation index, an admixture of reactive T-cells, residual follicular dendritic cell meshworks, and an indolent clinical course [1,5,10,27,28]. At the molecular level, PCFCL typically harbors alterations in *BCL11A* and *c-REL* genes, whereas PCDLBCL,LT is characterized by frequent chromosomal aberrations, particularly of chromosome 18, which includes amplifications of *BCL2* and *MALT1* genes [28,29,35,38].

Differentiating PCDLBCL,LT from other large BCL with secondary skin involvement, especially from extranodal DLBCL-NOS (not otherwise specified), the most frequent systemic large BCL to develop secondary skin involvement, is essential for accurate diagnosis, staging, and management [4,104]. While there may be morphologic overlap between PCDLBCL,LT and SC extranodal DLBCL-NOS, these entities display distinct clinical and molecular features [25].

Through GEP, two major subtypes of nodal DLBCL, which can rarely involve the skin, have been identified based on the cell of origin: the germinal center B-cell (GCB) subtype and the activated B-cell (ABC) subtype, each with distinct prognostic implications. Immunohistochemical algorithms, such as the Hans’ (and uncommonly, the Colomos’ and Muris’ algorithms), are used to classify these subtypes. GCB-type nodal DLBCL typically expresses CD10 and BCL-6, with variable BCL-2 positivity, while ABC-type nodal DLBCL consistently expresses MUM1 and FOXP1 [28,81,104,105]. PCDLBCL,LT shares immunophenotypic features with the ABC subtype [10,27,29,81,105]. However, PCDLBCL,LT often harbors *CD79B* mutations and the gain-of-function *MYD88* L265P mutation, which are less common both in nodal and extranodal DLBCL, particularly in the GCB subtype [81]. Additionally, PCDLBCL,LT frequently exhibits *BCL-2* gene rearrangements, gains in chromosome 18, and more commonly exhibits *MYC* rearrangements when compared to nodal DLBCL [30,89,90].

Other exceedingly rare differential diagnoses include intravascular DLBCL, plasmablastic lymphoma, anaplastic lymphoma kinase (ALK)-positive large BCL, and Burkitt lymphoma [3,10,27,28].

Ultimately, the diagnosis of PCDLBCL,LT and its differentiation from systemic or nodal large BCL with SC involvement requires a multidisciplinary approach to achieve accurate diagnosis and develop an appropriate, individualized treatment plan [1,5,6,10,28,38].

## 4. Conclusions and Future Directions

Primary cutaneous B-cell lymphomas encompass a diverse group of mature B-cell non-Hodgkin lymphomas, characterized by distinct clinical behaviors, histopathological features, and treatment challenges. The indolent subtypes, such as PCFCL and PCMZL/LPD, represent the majority of PCBCL cases and are typically associated with an excellent prognosis, albeit with a notable tendency for recurrence. Specifically, PCFCL is the most prevalent subtype, distinguished from systemic FL by the absence of *IGH::BCL2* translocations, which are characteristic of its systemic counterpart. This difference leads to a lack of BCL2 overexpression by immunohistochemistry in the majority of PCFCL cases. Molecular features such as somatic hypermutations and overexpression of oncogenic microRNAs are frequently observed. Despite exceptional survival outcomes, recurrences are common but generally do not affect long-term survival [1,4,5,6,28,38].

PCMZL/LPD, a neoplasm of post-germinal center B-cells, is often associated with chronic antigenic stimulation. Unlike PCFCL, PCMZL/LPD expresses BCL2 and lacks germinal center marker expression. Molecular abnormalities, including somatic hypermutations and mutations in the *FAS* gene, suggest impaired apoptotic regulation, contributing to disease persistence. Prognosis is excellent; however, cutaneous recurrences are frequent, and extracutaneous dissemination remains rare. Both PCFCL and PCMZL/LPD respond well to localized therapies, such as surgical excision and ISRT. Systemic therapies are typically reserved for disseminated or recurrent disease, with chemotherapy considered a last-line option due to its limited impact on long-term recurrence rates. Despite their indolent course, these lymphomas necessitate careful monitoring due to their recurrent nature, highlighting the need for continued research into optimal treatment strategies and the molecular mechanisms underlying relapse [1,4,5,6,28,38].

By contrast, PCDLBCL,LT represents a more aggressive entity, marked by frequent relapses and limited survival outcomes. Comprehensive diagnostic evaluation is essential for effective management. While the combination of R-CHOP plus ISRT remains the first-line treatment, novel agents such as lenalidomide and targeted therapies like ibrutinib show promise in relapsed and/or refractory cases [1,76,92,93,94].

Ongoing research into genetic markers and advanced monitoring techniques remains critical for improving disease management and patient outcomes in PCBCL. Key areas for further investigation include understanding the molecular mechanisms of treatment resistance, the interactions between the tumor microenvironment and immune evasion, and the molecular drivers of cutaneous homing and imprinting [106,107,108]. Future studies should focus on personalizing treatment regimens based on molecular profiles, optimizing the integration of targeted therapies with existing treatment protocols, and exploring immune-based therapeutic strategies. As with other rare diseases, multidisciplinary collaboration is essential to enhance diagnostic accuracy, disease monitoring, treatment paradigms, and long-term disease control, ultimately improving patient outcomes [28,38].

Furthermore, it is essential to continue including historically marginalized populations and People of the Global Majority (PoGM) in cutaneous lymphoma research. Efforts to ensure equitable representation will promote diversity and inclusion in medical research and clinical care, ultimately contributing to more comprehensive and globally relevant healthcare outcomes.

## Figures and Tables

**Figure 1 cancers-17-01202-f001:**
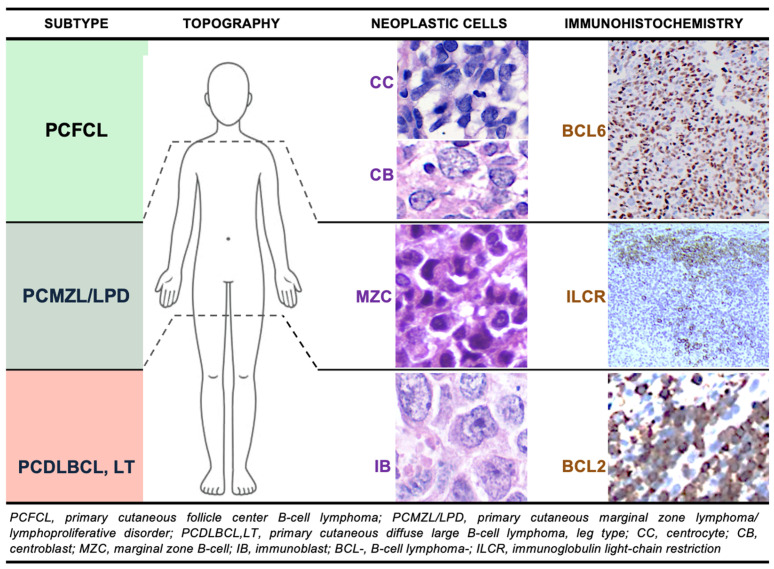
Archetypal features of indolent (green color) and aggressive (red color) primary cutaneous B-cell lymphomas/lymphoproliferative disorders. 400× magnification (neoplastic cells column); 100× for BCL6 and BCL2, and 40× for ILCR (immunohistochemistry column).

**Figure 2 cancers-17-01202-f002:**
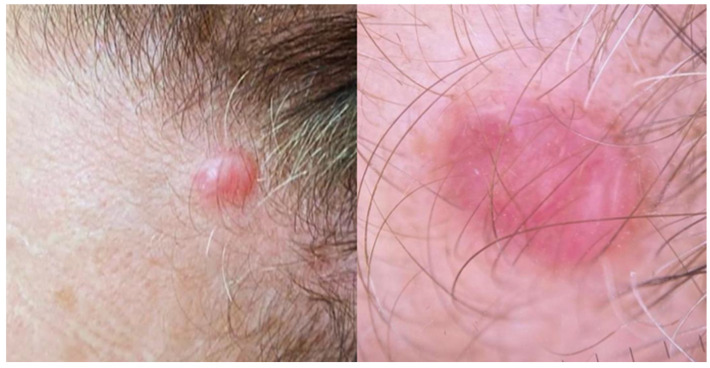
Primary cutaneous follicle center lymphoma presenting on the head. Clinical and dermoscopic (4×) images.

**Figure 3 cancers-17-01202-f003:**
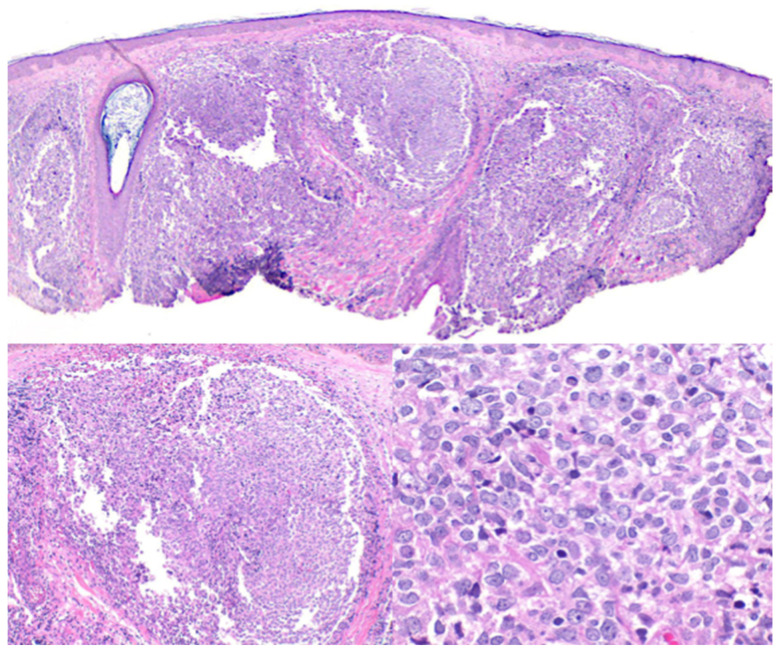
Primary cutaneous follicle center lymphoma. Hematoxylin and eosin-stained sections 20×, 40×, and 400×.

**Figure 4 cancers-17-01202-f004:**
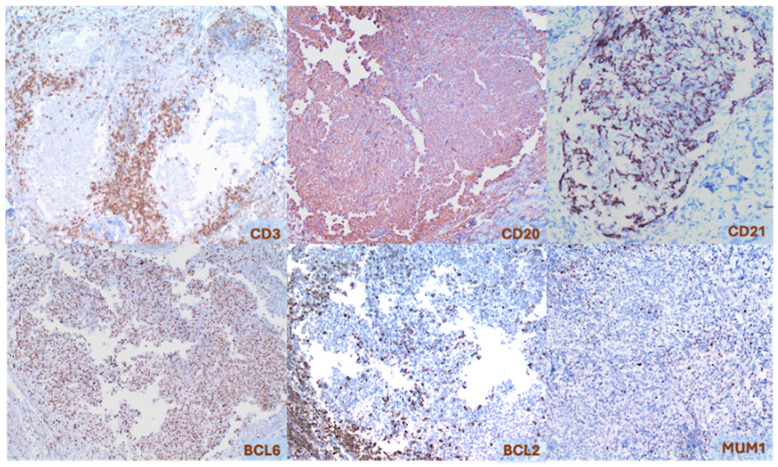
Primary cutaneous follicle center lymphoma. Immunohistochemistry, 40×.

**Figure 5 cancers-17-01202-f005:**
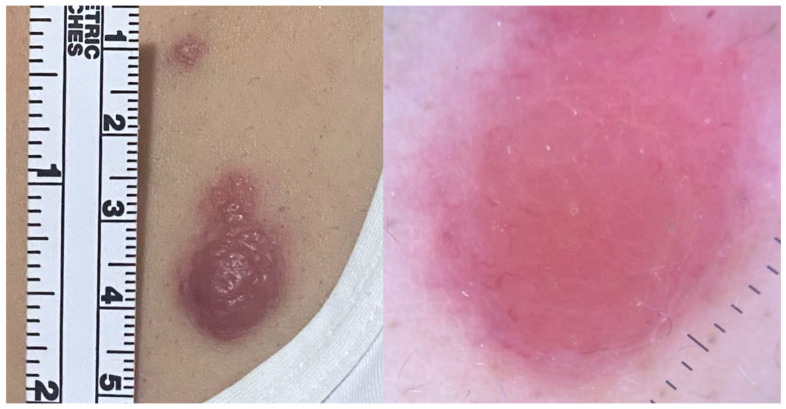
Primary cutaneous marginal zone lymphoma/lymphoproliferative disorder presenting on the trunk: clinical and dermoscopic (4×) images.

**Figure 6 cancers-17-01202-f006:**
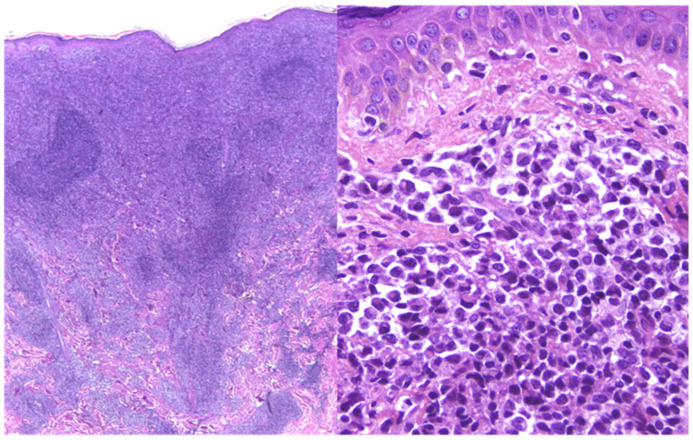
Primary cutaneous marginal zone lymphoma/lymphoproliferative disorder. Hematoxylin and eosin-stained sections 20× and 400×.

**Figure 7 cancers-17-01202-f007:**
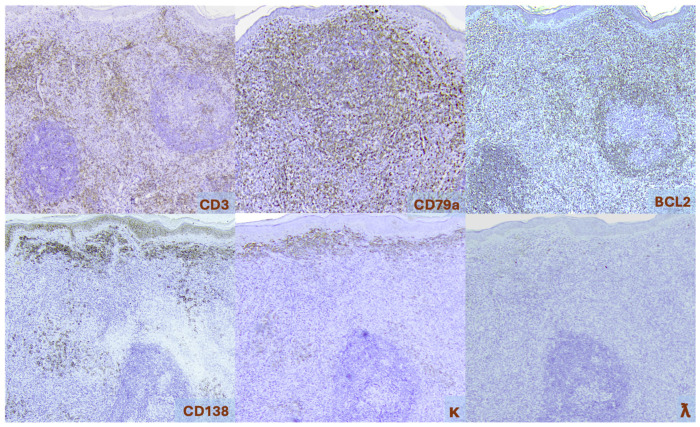
Primary cutaneous marginal zone lymphoma/lymphoproliferative disorder. Immunohistochemistry, 40×.

**Figure 8 cancers-17-01202-f008:**
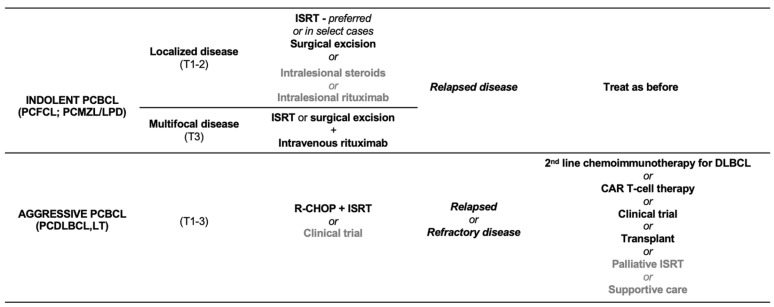
Simplified treatment approach for skin-limited indolent and aggressive primary cutaneous B-cell lymphomas/lymphoproliferative disorders.

**Figure 9 cancers-17-01202-f009:**
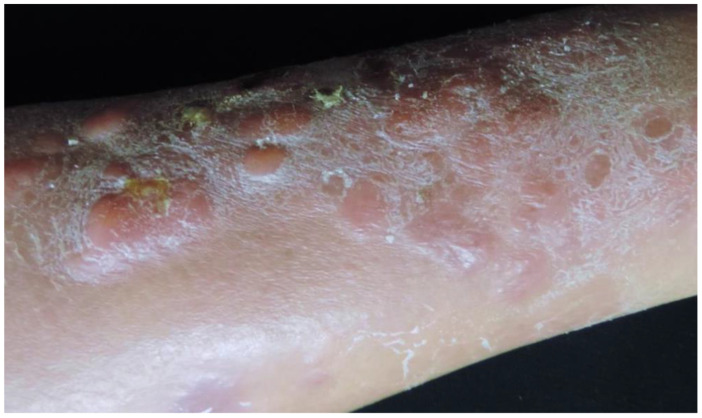
Primary cutaneous diffuse large B-cell lymphoma, leg type. Clinical image.

**Figure 10 cancers-17-01202-f010:**
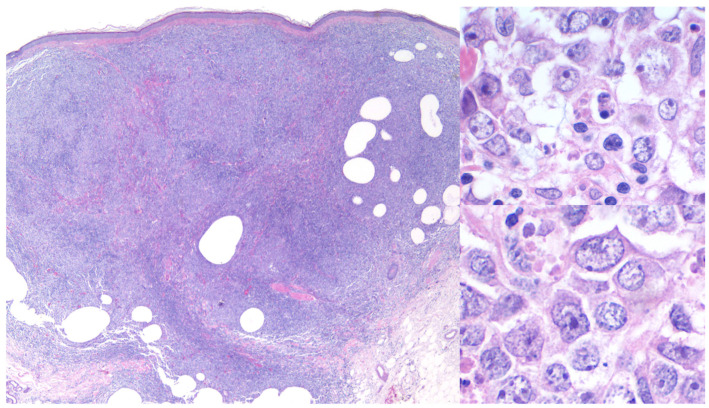
Primary cutaneous diffuse large B-cell lymphoma, leg type. Hematoxylin and eosin-stained sections 20×, 400× and 600×.

**Figure 11 cancers-17-01202-f011:**
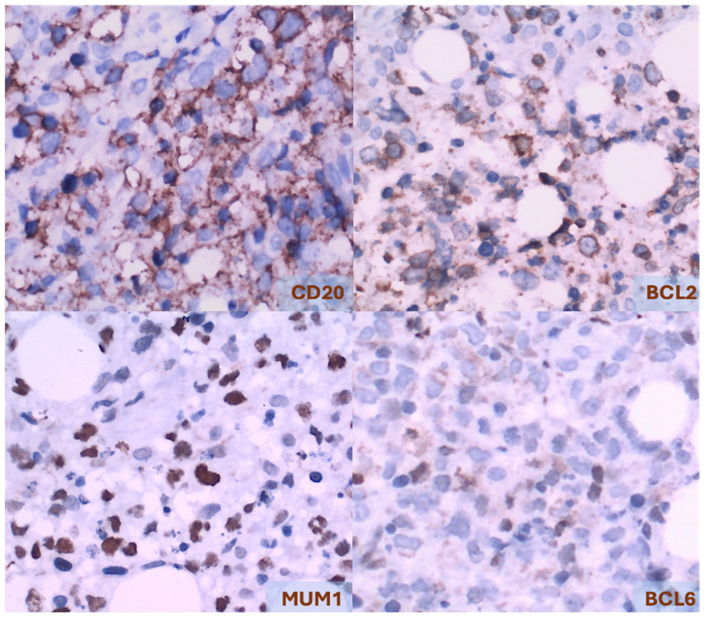
Primary cutaneous diffuse large B-cell lymphoma, leg type. Immunohistochemistry, 400×.

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
