# Peer review of "Unveiling Primary Cutaneous B-Cell Lymphomas: New Insights into Diagnosis and Treatment Strategies"

_cancers, 2025, doi:10.3390/cancers17071202_

Round 1

Reviewer 1 Report

Comments and Suggestions for Authors

This review is a comprehensive overview of  cutaneous B cell lymphoma, that focuses on histopathological and molecular aspects, with some mention of clinical  aspects. However, there are some inaccuracies, concerning some statements on systemic lymphomas. In particular:

  • on page 15 line 3 is reported that "nodal DLBCL often demonstrates MYC rearrangements". This is not correct because the Myc rearrangement is present in no more than 20% of systemic DLBCL. The cited reference, number 85, reports a relative high frequency of c-myc rearrangements in primary cutaneous DLBCL-NOS in comparison to DLBCL-LT, not to mention systemic lymphomas.
  •  on page 14 is reported that "Hans’, Colomos’ and Muris’ algorithms, are
    commonly used...". This is true only for Hans algorithm. Colomo and Muris are almost never used.
  • on page 13 is reported, concerning DLBCL-LT, that "MYC rearrangements are associated with poorer DSS and disease-free survival rates" without any reference. The authors should report the source of this statement, because they also report that double expressors and double hit  have no significant impact on prognosis.

The authors should discuss and adjust these three points.

Further minor notes are:

  • a typo on page 2 line 25: all instead of al
  • on page 3 the acronym FCBCL should be explained

Author Response

Comments 1: This review is a comprehensive overview of cutaneous B cell lymphoma, that focuses on histopathological and molecular aspects, with some mention of clinical  aspects. However, there are some inaccuracies, concerning some statements on systemic lymphomas. In particular: on page 15 line 3 is reported that "nodal DLBCL often demonstrates MYC rearrangements". This is not correct because the Myc rearrangement is present in no more than 20% of systemic DLBCL. The cited reference, number 85, reports a relative high frequency of c-myc rearrangements in primary cutaneous DLBCL-NOS in comparison to DLBCL-LT, not to mention systemic lymphomas.

Response 1: Thank you for pointing this out. We agree with the overall and first specific comment. Therefore we have now focused the core of the manuscript on clinical aspects and therapy and also corrected the statement regarding MYC rearrangements, in keeping with the reviewers request (see page 13, lines 482 to 484)

Comments 2: on page 14 is reported that "Hans’, Colomos’ and Muris’ algorithms, are
commonly used...". This is true only for Hans algorithm. Colomo and Muris are almost never used.

Response 2: Thank you for pointing this out. We agree with this comment. Therefore we have now altered this statement, in keeping with the reviewers request (see page 13, lines 475 to 477)

Comments 3: on page 13 it is reported, concerning DLBCL-LT, that "MYC rearrangements are associated with poorer DSS and disease-free survival rates" without any reference. The authors should report the source of this statement, because they also report that double expressors and double hit have no significant impact on prognosis.

Response 3: Thank you for pointing this out. We agree with this comment. Therefore we have now reported the source of this statement, in keeping with the reviewers request (see page 12, lines 405 to 408)

Comments 4 & 5: Further minor notes are: a typo on page 2 line 25: all instead of al; on page 3 the acronym FCBCL should be explained

Response 4: Thank you for pointing this out. We have corrected the typo (page 2, line 77) and changed the abbreviation FCBCL for PCFCL, in keeping with the rest of the manuscript (page 4, line 139)

Reviewer 2 Report

Comments and Suggestions for Authors

This manuscript provides a review on primary cutaneous B-cell lymphomas, including its pathology and treatment strategies. There are some concerns.

(1) The review on primary cutaneous B-cell lymphomas has been reported before (for example, Diagnosis and Treatment of Primary Cutaneous B-Cell Lymphomas: State of the Art and Perspectives, Cancers, 2020, 12(6):1497, doi: 10.3390/cancers12061497; Primary Cutaneous B-Cell Lymphomas, Surgical Pathology Clinics, 2014, 7, 253, doi: 10.1016/j.path.2014.02.002; Primary Cutaneous B-Cell Lymphomas: An Update, Front Oncol, 2020, 10, 651, doi: 10.3389/fonc.2020.00651; Primary cutaneous B-cell lymphoma: review and current concepts, J Clin Oncol, 2000, 18, 2152, doi: 10.1200/JCO.2000.18.10.2152). In this case, the unique points of this review should be illustrated in more details.

(2) Besides providing the cases, it might also be worthy to provide a table to summarize the current diagnostic markers or treatment strategies in different stages, which might be interesting to readers.

(3) It might also be worthy to summarize the pathology of this primary cutaneous B-cell lymphomas in a figure, which might provide a clear picture of this disease.

(4) The treatment approach of primary cutaneous B-cell lymphomas should also be summarized.

Author Response

Comments: This manuscript provides a review on primary cutaneous B-cell lymphomas, including its pathology and treatment strategies. There are some concerns.

Comment 1: The review on primary cutaneous B-cell lymphomas has been reported before (for example, Diagnosis and Treatment of Primary Cutaneous B-Cell Lymphomas: State of the Art and Perspectives, Cancers, 2020, 12(6):1497, doi: 10.3390/cancers12061497; Primary Cutaneous B-Cell Lymphomas, Surgical Pathology Clinics, 2014, 7, 253, doi: 10.1016/j.path.2014.02.002; Primary Cutaneous B-Cell Lymphomas: An Update, Front Oncol, 2020, 10, 651, doi: 10.3389/fonc.2020.00651; Primary cutaneous B-cell lymphoma: review and current concepts, J Clin Oncol, 2000, 18, 2152, doi: 10.1200/JCO.2000.18.10.2152). In this case, the unique points of this review should be illustrated in more details.

Response 1: Thank you for pointing this out. In keeping with your request, we have illustrated in more detail the unique point of our review. Notably we have highlighted the changes in the classification of primary cutaneous lymphomas since the 2022 iterations of the WHO and ICC, new insights into the molecular pathogenesis of disease, and importantly, changes in the treatment paradigm of these lymphoid neoplasms. Such important points are based on recent publications, mostly dating after 2020 (the last date when the afore mentioned reviews were published). We are hope that the changes we further made to our new manuscript adequately reflect the unique points of our updated review.. 

Comments 2: Besides providing the cases, it might also be worthy to provide a table to summarize the current diagnostic markers or treatment strategies in different stages, which might be interesting to readers.

Response 2: Thank you for pointing this out. We agree with this and thus have created a new flow chart figure that summarizes the treatment strategies of PCBCL according to stage (see figure 8, page 9)

Comments 3:  It might also be worthy to summarize the pathology of this primary cutaneous B-cell lymphomas in a figure, which might provide a clear picture of this disease.

Response 3:  Thank you for pointing this out. We agree with this and thus have created a new figure summarizing the salient points of the clinical, histopathological and immunohistochenical features of PCL (see figure 1, page 3)

Comments 4: The treatment approach of primary cutaneous B-cell lymphomas should also be summarized.

Response 4: Thank you for pointing this out. We agree with this and thus have created a new flow chart figure that summarizes the treatment strategies of PCBCL (see figure 8, page 9)

Reviewer 3 Report

Comments and Suggestions for Authors

This review summarizes the clinical, pathological, and molecular characteristics, treatment strategies, and the differential diagnosis of three representative cutaneous B-cell lymphomas (CBCLs), primary cutaneous follicle center lymphoma (PCFCL), primary cutaneous marginal zone lymphoma/lymphoproliferative disorder (PCMZL/LPD) and primary cutaneous diffuse large B-cell lymphoma, leg type (PCDLBCL, LT). This paper is well-written and well-organized. I have some minor comments below.

  1. The prevalence of PCFCL may vary by race. Although most common in Caucasians, PCFCL is the least CBCL in Asia based on the data from Japan and Korea (Fujii K et al., J Dermatol Sci, 2020; 97: 187-193; Moon IJ et al., Sci Rep, 2024; 14: 20118).

  1. The addition of BCL2 and CD21 staining in Figure 3 is recommended.

  1. The addition of CD79a and BCL2 staining in Figure 6 is recommended.

  1. Some reports suggest that dual protein expression of BCL2 and MYC, MYC rearrangements, loss of CDKN2A, and the somatic MYD88 L265P mutation negatively impacted on prognosis in PCDLBCL, LT (Menguy S, et al., Mod Pathol, 2018; 31: 1332-1342; Schrader AMR et al., Am J Surg Pathol, 2018; 42: 1488-1494: Senff NJ, et al., J Invest Dermatol 2009; 129: 1149-1155; Pham-Ledard A, et al., JAMA Dermatol 2014; 150: 1173-1179).

  1. Immunohistochemical labels of Figure 3, 6, and 9, which are presented in the PDF file named original image, are missing in the manuscript file.

Author Response

Comments: This review summarizes the clinical, pathological, and molecular characteristics, treatment strategies, and the differential diagnosis of three representative cutaneous B-cell lymphomas (CBCLs), primary cutaneous follicle center lymphoma (PCFCL), primary cutaneous marginal zone lymphoma/lymphoproliferative disorder (PCMZL/LPD) and primary cutaneous diffuse large B-cell lymphoma, leg type (PCDLBCL, LT). This paper is well-written and well-organized. I have some minor comments below.

Comments 1: The prevalence of PCFCL may vary by race. Although most common in Caucasians, PCFCL is the least CBCL in Asia based on the data from Japan and Korea (Fujii K et al., J Dermatol Sci, 2020; 97: 187-193; Moon IJ et al., Sci Rep, 2024; 14: 20118).

Response 1: Thank you for pointing this out. We agree with this. Therefore we have now added a statement recognizing that the prevalence of PCFCL may vary by race, and both elaborated and cited the work from the aformentioned studies (page 3, lines 107-111)

Comments 2: The addition of BCL2 and CD21 staining in Figure 3 is recommended. 

Response 2: Thank you for pointing this out, we agree. Therefore we have included the suggested immunohistochemical stains to the figure (now numbered figure 4 on page 5)

Comments 3: The addition of CD79a and BCL2 staining in Figure 6 is recommended.

Response 3: Thank you for pointing this out, we agree. Therefore we have included the suggested immunohistochemical stains to the figure (now numbered figure 7 on page 7)

Comments 4: Some reports suggest that dual protein expression of BCL2 and MYC, MYC rearrangements, loss of CDKN2A, and the somatic MYD88 L265P mutation negatively impacted on prognosis in PCDLBCL, LT (Menguy S, et al., Mod Pathol, 2018; 31: 1332-1342; Schrader AMR et al., Am J Surg Pathol, 2018; 42: 1488-1494: Senff NJ, et al., J Invest Dermatol 2009; 129: 1149-1155; Pham-Ledard A, et al., JAMA Dermatol 2014; 150: 1173-1179).

Response 4: Thank you for pointing this out. We agree. Therefore we have elaborated on the matter, in keeping with the reviewers suggestion (page 13, lines 401-410)

Comments 5: Immunohistochemical labels of Figure 3, 6, and 9, which are presented in the PDF file named original image, are missing in the manuscript file

Response 5: Thank you for pointing this out. We have added the missing immunohistochemical labels to the new version of the manuscript file

Round 2

Reviewer 2 Report

Comments and Suggestions for Authors

This revised version can be acceptable.

Author Response

Comments 1: This revised version can be acceptable

Response 1: We thank the reviewer for their constructive comments throughout the process of peer review and are pleased to know they found this revised version acceptable. 

Reviewer 3 Report

Comments and Suggestions for Authors

The authors revised the manuscript well, but I have still one minor concern, which was not responded by the authors.

It will be better to add CD79a staining in Figure 7, because CD20 staining seemed to be scarce as PCMZL/LPD.

Author Response

Comment 1: The authors revised the manuscript well, but I have still one minor concern, which was not responded by the authors. It will be better to add CD79a staining in Figure 7, because CD20 staining seemed to be scarce as PCMZL/LPD.

Response 1: Thank you for pointing this out. We agree with the reviewer and admit that I (the corresponding author) misread the original request by the reviewer (and mistakenly thought they were requesting a CD138 immunostain). CD79a IHC was not pursued as part of the original analysis of this case. However, we retrieved the block and whilst very little tissue was left, we have now replaced the CD20 image with one from the requested CD79a immunostain. Unfortunately the exact area of the biopsy captured in the pictures for the other immunostains was no longer within the tissue. However, we hope this alteration is acceptable and satisfies the reviewer's keen request.